

# Stage 2 registered report: investigating a preference for certainty in conversation among autistic adults

Alexander C. Wilson and Dorothy V.M. Bishop

Department of Experimental Psychology, University of Oxford, Oxford, United Kingdom

Corresponding author
Alexander C. Wilson,
alexander.wilson2@psy.ox.ac.uk

## ABSTRACT

Social communication difficulties are a diagnostic feature in autism. These difficulties are sometimes attributed, at least in part, to impaired ability in making inferences about what other people mean. In this registered report, we tested a competing hypothesis that the communication profile of adults on the autism spectrum can be more strongly characterised by reduced confidence in making inferences in the face of uncertain information. We tested this hypothesis by comparing the performance of 102 autistic and 109 non-autistic adults on a test of implied meaning, using a test of grammaticality judgements as a control task. We hypothesised that autistic adults would report substantially lower confidence, allowing for differences in accuracy, than non-autistic adults on the test of implied meaning compared to the grammaticality test. However, our results did not suggest this. Instead, we found that accuracy and confidence were both reduced to a similar extent on the test of implied meaning in the autistic group compared to the control group, although these were only subtle differences. This pattern of results was specific to inference-making, as the autistic and non-autistic groups did not differ on the grammar test. This supports the idea that specific differences in pragmatic language processing can exist in autism in the absence of core language problems. Importantly, this pattern of results (differences on the test of implied meaning and no differences on the grammar test) was reversed in a group with self-reported reading difficulties, indicating that the differences in inference-making were specific to the autistic group. Lastly, we found relationships between Intolerance of Uncertainty, performance on the test of implied meaning, and self-reported social communication challenges. This supports the idea that discomfort with uncertainty plays a role in the pragmatic language and communication challenges in autism.

## INTRODUCTION

Persistent challenges with social communication are a defining feature for the diagnosis of autism (*American Psychiatric Association, 2013*). The underlying nature of these challenges remains unclear, although they are sometimes attributed to a core impairment in pragmatics (*e.g., Baron-Cohen, 1988*; *Rapin & Dunn, 2003*). Pragmatics refers to the role of context in communication, including the ability to "read between the lines" to infer intended meaning beyond what is explicitly stated (*Baird & Norbury, 2016*). However, empirical

research suggests that pragmatic difficulties may be rather subtle in autistic people, and mostly attributable to language ability (*Kalandadze et al., 2018*; *Loukusa & Moilanen, 2009*). An alternative suggestion is that social communication difficulties are less the result of an impairment in pragmatics, but more impacted by cognitive preferences that differ between autistic and non-autistic people. We propose that a preference for certainty and explicit communication commonly occurs in autistic people, and that this trait may be a critical factor in the communication difficulties experienced by autistic people, as communicative situations often involve ambiguity, uncertainty and implied meanings.

## Intolerance of uncertainty

A "preference for certainty and explicit communication" may link to the widely-researched cognitive trait, Intolerance of Uncertainty (IU). IU has been defined as a tendency to negatively evaluate uncertain situations and information (*Shihata et al., 2016*). We use the term "Intolerance of Uncertainty" in line with previous research and intend to convey a value-neutral meaning in using it, as we recognise that high levels of IU may be an understandable, even adaptive, response where individuals have experienced mishaps in confusing situations. IU has mostly been investigated as a transdiagnostic construct that plays a central role in emotional disorders across the general population (see *Shihata et al., 2016* for a review), but it also seems especially relevant to autism, with autistic children and adults showing significantly elevated levels of the trait compared to the general population (*e.g.*, *Hwang et al., 2020*; *Vasa et al., 2018*). IU has been closely linked to anxiety in autistic people (*Jenkinson, Milne & Thompson, 2020*), and also relates to core features of autism, including social difficulties, sensory sensitivities, insistence on sameness and repetitive behaviours (*e.g.*, *Hwang et al., 2020*; *Vasa et al., 2018*; *Wigham et al., 2015*).

A possible link between IU and communication in autistic people remains largely unexplored. However, there are reasons to believe that a link is plausible. First, inferential models of communication, such as Relevance Theory (*Sperber & Wilson, 1986*), propose that communication inherently involves uncertainty. Under Relevance Theory, language comprehension is not simply a process of "understanding what the words mean", as there are often indeterminacies and ambiguities in uses of language; instead, the words are used as evidence by the listener in supporting a hypothesis about what the speaker *probably* means in the context, *i.e.*, inferring intended meaning under uncertainty. Relevance Theory suggests that there is a gradient of uncertainty in communication. Sometimes, the listener can rely mostly on the explicit content of the utterance to compute the intended meaning, but in other situations there is a greater reliance on inferential processing to understand the speaker's probable meaning by integrating the utterance with contextual cues and world knowledge. Compare for instance the utterances "No, let's stay inside" and "It's quite cold today" as responses to a suggestion to go outside. In the second example, the speaker communicates implicitly, leaving the listener to process the implicature that they would probably prefer to stay inside. In a previous study, we provided evidence for cognitive differences between autistic and non-autistic people in processing implicatures (*Wilson & Bishop, 2020c*). Crucially, it seemed that a cognitive preference for certainty and explicit communication, and not simply reduced ability, may account for some of the differences.

Participants completed the Implicature Comprehension Test, which required individuals to listen to short conversational interchanges that are followed by a comprehension question to assess whether an implied meaning has been processed; test-takers responded with "yes", "no" or "don't know". Controlling for grammar/vocabulary ability, we found that autistic adults ($N = 66$) were 6.19, 95% CI [3.63–10.39], times more likely to select the "don't know" rather than the correct response compared to non-autistic people ($N = 118$), and also 2.56, 95% CI [1.76–3.77], times more likely to choose the "incorrect" response (*Wilson & Bishop, 2020c*). Group differences were large, and performance on the test gave 76% sensitivity and specificity for differentiating between autistic and non-autistic groups. On the face of it, these results suggest that autistic people have difficulties inferring the gist of a speaker's meaning, as predicted by the 'central coherence' theory, which proposes that autistic people may show less tendency than non-autistic people to process information at a global level (*Frith, 1989*).

However, in an alternative version of the test without a "don't know" response, autistic individuals showed high accuracy for items for which they had selected "don't know" first time round. This marked tendency to select "don't know" when given a chance, but to process the inference as intended when constrained by the task, suggested reduced confidence in the face of uncertain information and a preference for explicit communication. This could be due to possible difficulties around metacognition in autistic people, who may experience a mismatch between performance and confidence in their performance due to differences in self-monitoring. There is evidence that autistic people may show such a mismatch (*Grainger, Williams & Lind, 2016*; *Nicholson et al., 2019*) although there is some concern about the replicability of these results (*Maras, Norris & Brewer, 2020*). An alternative view would be that it is less an issue of metacognitive "ability", and more about differences in personality/cognitive preference, with the well-replicated elevated levels of IU in autistic people (*e.g.*, *Hwang et al., 2020*; *Vasa et al., 2018*) accounting for this apparent preference for explicit communication observed in our previous study. In the present study, we aim to replicate this finding with more refined methods. In an adapted version of the Implicature Comprehension Test, individuals will respond using a 4-point scale of "yes", "maybe yes", "maybe no", and "no", allowing us to capture accuracy and confidence in the same measure. We hypothesise that confidence is likely to be affected specifically in a pragmatic language task (*i.e.*, where the individual needs to make flexible context-dependent inference about uncertain implied meanings), and not on tasks focused on more structural, codified aspects of language such as grammatical competence. As such, we present another metalinguistic task, the Grammaticality Decision Test, as a control task with a similar response format to the pragmatic task to test the specificity of any differences.

We propose that reduced confidence on the Implicature Comprehension Test will be a marker of IU in autistic people, and may be a more influential factor in the communication difficulties diagnostic of autism, as opposed to a "deficit" in understanding social meanings. If this claim is borne out, it would have a couple of implications for psychological practice. First, it would suggest that interventions targeting IU may be useful for autistic people wanting support with communication challenges. Current interventions

for communication focus on explicit instruction in social skills, and reviews suggest modest effectiveness although there are questions about the extent to which skills transfer to daily life (*Gates, Kang & Lerner, 2017*; *Spain & Blainey, 2015*). A focus on IU may be a useful alternative target. Existing cognitive interventions involve integrating psychoeducation and cognitive challenge techniques to target a client's beliefs about (un)certainty, and these have shown some effectiveness for treating mental health difficulties and particularly anxiety in the general population (*Shihata et al., 2016*). It remains to be seen whether such interventions could be adapted to support autistic individuals with distressing communication experiences, although this is a promising possibility given that early studies suggest that such interventions may be feasible and acceptable for autistic groups (*Rodgers et al., 2018*; *Rodgers et al., 2017*). Second, if a cognitive preference for certainty is especially significant as an explanation for social difficulties, then it supports an autism-positive approach to intervention which focuses on awareness of cognitive differences across communities. In addition, if performance on the Implicature Comprehension Test is a sensitive marker of IU, that in itself might have clinical and research utility, since measurement of IU is currently limited to self- and informant-report questionnaires.

A remaining question is whether any differences observed on our tasks are specific to autism or might also be relevant to other neurodevelopmental diagnoses. This is certainly plausible in the light of dimensional models of neurodiversity, where features of autism, developmental language disorder, dyslexia, ADHD, etc., might show some overlap and exist as a continuum in the general population (*Thapar, Cooper & Rutter, 2017*). To test the specificity of any cognitive differences observed on our tests, we will compare performance by autistic people to both a dyslexic and a general population sample. As neurodevelopmental conditions are often co-occurring, we view these three groups as defined less by a specific diagnostic label but rather as varying along a communication continuum. As such, one group is defined by social communication differences potentially alongside co-occurring language/literacy impairments (the autism group), a second group by language/literacy impairments but no diagnosed social communication difficulties (the dyslexia group), and a final group without any communication, language or literacy related diagnosis. It is possible that dyslexic adults may show some difficulty on our pragmatic task (*i.e.*, the Implicature Comprehension Test), as previous research has documented some limited evidence for pragmatic difficulties in dyslexic individuals (*e.g.*, *Cappelli et al., 2018*; *Cardillo et al., 2018*; *Griffiths, 2007*). An alternative possibility is that adults with dyslexia will show greater difficulty with tasks focused more on structural language skills compared to pragmatics. For instance, a meta-analysis has found that dyslexic adults perform less well on language measures, such as vocabulary, speeded naming, verbal memory and phonological processing, than people without a diagnosis of dyslexia, with moderate to large effect sizes (*Swanson & Hsieh, 2009*). Given that there is no clear reason to support one of these possibilities over the other, we will take a more exploratory approach with the dyslexic group to examine how they compare with autistic adults.

In summary, we propose the following hypotheses:

(1) Autistic adults will score lower on a pragmatic language task when responses are coded purely in terms of confidence (number of yes and no responses, regardless of polarity)

than when responses are coded in terms of accuracy (with yes and maybe yes, and maybe no and no responses, combined according to polarity), compared to adults without any neurodevelopmental diagnosis, but will not show this same disparity between accuracy and confidence on a core language task.

(2) The number of less confident responses (maybe responses) on the pragmatic language task, the score on the Intolerance of Uncertainty Scale, and self-reported social communication difficulties will significantly intercorrelate across the full sample.

## METHODS

Methods and our analysis plan for this project were published as a registered report (*Wilson & Bishop, 2020b*): https://peerj.com/articles/10398/. Ethical approval for this project was granted on 30/03/2020 by the Medical Science Interdivisional Research Ethics Committee at Oxford University (Ref: R68518/RE001). Scripts, data and example materials for this project are available on the Open Science Framework here: https://osf.io/wk97s/. We are very happy to share full materials for this study, but to protect the validity of the items for future uses, we ask that researchers contact us to request a link to the full assessments. See further information on requesting access by following the link above.

### Power calculation

We determined power to detect the three-way interaction described in Hypothesis 1 using simulations. We used data reported in *Wilson & Bishop (2020c)* to estimate the likely size of fixed and random effects in the mixed model described in Data Analysis below. Using R package simr (*Green & MacLeod, 2016*) we ran 1,000 simulations with a sample size of 200 people (100 autistic, 100 non-autistic) and a significant three-way interaction was found in 9,830 simulations, indicating that power was over 98% to detect our effect of interest at an alpha level of .05. Effectively, this allowed us to detect a significant difference where approximate Cohen's $d$ values in favour of the non-autistic group were 0.70 and 1.10 for the implicature accuracy and confidence variables and 0.20 for the grammar variables, as suggested by our previous data. Allowing for exclusion of up to 10% of participants during the outlier exclusion phase described in Data Analysis, power remained very high (98% in a sample of 180). For Hypothesis 2, a sample of 200 was powered at over 99% to detect a correlation of .3.

### Participants

We recruited individuals with autism, self-reported reading difficulties, and no neurodevelopmental diagnosis. Based on the power calculation, we aimed to recruit 100 autistic adults and 100 adults without a neurodevelopmental diagnosis in order to run the confirmatory analysis. In addition, we set out to recruit 50 dyslexic adults as a clinical control group for exploratory analysis. All participants met the following eligibility criteria: (i) age of 18 years or over, (ii) native-level fluency in English, (iii) no history of acquired brain injury, (iv) no significant uncorrected sensory impairment, and (v) access to a computer with internet and audio.

Individuals were recruited into three groups defined by communication and language/literacy problems. One group was recruited on the basis of a clinical diagnosis of

autism; participants needed to declare where, by whom and what label was used for their diagnosis on the Study Questionnaire. For inclusion, the diagnosis must have been made in a clinical service by appropriately trained individuals, such as clinical psychologists, psychiatrists or developmental paediatricians. We recruited autistic individuals through Autistica, the research network for families and individuals with autism, as well as support groups arranged privately and by the National Autistic Society, and through social media. A second group included individuals reporting dyslexia or specific reading difficulties. For inclusion in this group, individuals needed to score below the clinical threshold of 6 on the Autism Spectrum Quotient (AQ) and at 6 or above on the Reading Scale of the Adult Reading Questionnaire (ARQ); this latter score translates to over 1.5 SDs above the mean in individuals not self-reporting dyslexia in the original validation study (*Snowling et al., 2012*). Individuals were recruited through charitable organisations such as the British Dyslexia Association and social media. Other neurodevelopmental diagnoses were not grounds for exclusion from these groups. A third group had no neurodevelopmental diagnosis, and was recruited via the online participant platform, Prolific (https://prolific.co). Individuals were excluded from this third group if they scored above threshold on either the AQ or ARQ (*i.e.*, 6 or above on either) or if they have ever been diagnosed with: a global or specific learning disability, attention deficit hyperactivity disorder, dyspraxia/developmental coordination disorder, a genetic variation (such as Down's syndrome or Fragile-X) or a neurological condition (such as epilepsy).

## Procedure

The study was presented online using Gorilla, the online platform for behavioural experiments and surveys (https://gorilla.sc/). Individuals completed an online set of tasks and questionnaires in one sitting at a time and place of their choosing. After providing informed written consent to participate, individuals completed a Study Questionnaire (please see the OSF link) on which they reported on demographics and any neurodevelopmental diagnoses. Then they completed questionnaires/tasks in two sections. The first section included the experimental tasks required for the hypothesis-testing, and the second included several brief measures for characterising the sample. The two experimental tasks were randomized between participants, and all other measures were administered in the order set out below.

## Measures
### Section 1: Experimental tasks assessing ability and confidence with pragmatics and core language

*Implicature Comprehension Test-2 (ICT-2).* In this test of pragmatic language comprehension, participants completed an adapted version of the Implicature Comprehension Test (*Wilson & Bishop, 2019*). There was a sequence of 56 videos, each approximately 8 s long, consisting of a conversational adjacency pair between two characters: the first character asked a closed question (eliciting a "yes" or "no" answer) and the second character produced a short answer but did not say yes or no. Each utterance was between 5 and 10 words in length and grammatically simple, and age of acquisition of the words did not exceed middle primary school level. Following the dialogue, the participant heard a comprehension question directly based on the structure of the first character's question.

The participant answered the question on a 4-point scale (yes, maybe yes, maybe no, no) by clicking buttons arranged horizontally on the screen. This was a timed task, with a time limit of 10 s for a response from the offset of the question. There were two item types: implicature and explicit-response. Utterance length and psycholinguistic variables (word frequency, word age-of-acquisition and word concreteness) were controlled across the two item types.

For 40 videos, the second character's answer was indirect, and the participant needed to process implicature to answer the comprehension question appropriately. Example:

Character 1: Did you hear what the police said?

Character 2: There were lots of trains going past.

Comprehension Question: Did he hear what the police said?

Answer: No

Half of the comprehension questions were correctly answered by "yes" and half by "no". There were two measured variables: total accuracy (collapsing yes and maybe yes, and maybe no and no responses, according to polarity) out of 40 and total confidence (number of yes and no responses, regardless of polarity) out of 40.

Alongside the implicature items, there were 16 explicit-response items where the second character's answer is more explicit. In these items, the speaker intended to convey uncertainty explicitly, whereas in the implicature items, the uncertainty was in the mind of the listener. Example:

Character 1: Will we get there by seven?

Character 2: Mmm, yes maybe, I think we're near.

Comprehension Question: Will they get there by seven?

Answer: Maybe yes

For these items, the comprehension questions encouraged the participant to use the full scale, with four questions each correctly answered by "yes", "maybe yes", "maybe no" and "no". There was one measured variable: total accuracy out of 16.

*Grammaticality Decision Test (GDT; based on Wilson & Bishop, 2019).* In this test of core language ability, participants listened to a sequence of 50 sentences and decided if the sentence was grammatical and well-formed or not. Half the sentences were grammatical. Grammatical violations represented mistakes that native speakers would not tend to make, such as using an incorrect verb form (*e.g.*, I went out after I have eaten dinner) or atypical placing of adverbs (*e.g.*, If you can't find it, I can send again the letter). Participants were asked whether the sentences were grammatical, indicating "yes", "maybe yes", "maybe no" and "no" as their answer by clicking buttons arranged horizontally on the screen, as in the ICT-2. After offset of the sentence, participants had 10 s to give their response.

### Section 2: Questionnaires and tasks for characterising the sample

*Autism Spectrum Quotient-10 (AQ-10; Allison, Auyeung & Baron-Cohen, 2012).* Autistic traits were measured using this 10-item version of the Autism Spectrum Quotient (AQ). In the original validation study, the measure had 85% correct discrimination between almost 450 autistic adults and over 800 control adults. The National Institute for Health and Care

Excellence (*National Institute for Health and Care Excellence, 2012*) recommend use of the questionnaire for identifying individuals for comprehensive autism assessment. A clinical cut-off of 6 or more is taken as indicating possible autism.

*Communication Checklist Self-Report (CC-SR; Bishop, Whitehouse & Sharp, 2009).* This is a norm-referenced questionnaire measuring self-reported communication challenges. Participants were presented with the pragmatic language scale (22 items). For each item, participants identified how frequently certain communication behaviours apply to them on a 4-point scale from "less than once a week (or never)" to "several times a day (or all the time)". An example item is "People tell me that I ask the same question over and over". *The authors had permission to use this instrument from the copyright holders.*

*Adult Reading Questionnaire (ARQ) reading scale (Snowling et al., 2012).* Self-reported reading difficulties were measured using this 5-item questionnaire. In the original validation study, it showed good construct validity (correlating with observed literacy ability at $-.67$) and, along with self-reported dyslexia status, discriminated with 88% accuracy in identifying those with weaker literacy skills.

*International Cognitive Ability Resource (ICAR) sample test (Condon & Revelle, 2014).* This is an open-access test of general cognitive ability, which requires participants to complete 16 items across four item types: matrix reasoning, verbal reasoning, three-dimensional rotation, and letter-number sequences. In the original validation study, the ICAR sample test had good internal consistency (alpha = .81), and good convergent validity (correlating at approximately .8 with commercial IQ measures when correcting for reliability and restriction of range; *Condon & Revelle, 2014*; *Young & Keith, 2020*).

*Synonyms test (Wilson & Bishop, 2019).* General verbal ability was measured using this 25-item test of vocabulary knowledge. Participants selected which of five written words is synonymous with a target word, under a 12-second time limit. The original version of the GDT and this task showed a moderate correlation in both autistic and non-autistic samples, suggesting they are overlapping measures of core language ability (*Wilson & Bishop, 2019*; *Wilson & Bishop, 2020a*).

*Intolerance of uncertainty scale (IUS-12; Carleton, Norton & Asmundson, 2007).* In this self-report measure of intolerance of uncertainty, participants were presented with 12 statements about uncertainty, ambiguous situations, and the future. They rated how closely each statement relates to them on a 5-point scale from "not at all characteristic of me" to "entirely characteristic of me". An example item is "When I am uncertain, I can't function very well".

## A priori data analysis plan

Individuals were excluded from the dataset if they had an outlying score for either (a) accuracy on the GDT or the positive control items of the ICT-2, or (b) total number of timeouts across the ICT-2 and GDT. Outliers were defined according to the method of *Hoaglin & Iglewicz (1987)*: more than 2.2 times the interquartile range below the

first quartile. In previous work, these criteria led to exclusion of approximately 5% of participants, and captured individuals scoring below approximately 50% on the GDT and 70% on the positive control items of the original version of the ICT (*Wilson & Bishop, 2020a*).

Data were analysed in R (*R Core Team, 2019*). After exclusions, total scores on the two experimental tasks for the groups with autism and no neurodevelopmental diagnosis were turned into long format, and each participant's total was dummy-coded for task (ICT-2 or GDT), group (autistic or no neurodevelopmental diagnosis), response (accuracy or confidence) and participant. We ran a mixed effects linear regression using the lme4 R package (*Bates et al., 2015*). The model included three fixed effects (task, group and response) and the interactions between these, as well as a random effect (participant). The significance level of the three-way interaction offered a test of Hypothesis 1. We also computed correlations between confidence on the ICT-2, self-reported communication challenges on the CC-SR and total score on IUS-12 across the full sample; this tested Hypothesis 2 and represented a dimensional analysis of the relationship between communication difficulties and sensitivity to uncertainty. Table 1 shows a summary of our planned analyses, linking research questions, hypotheses, tests and power calculations.

In exploratory analysis, we examined how the group with self-reported reading difficulties compared to the autistic group and the group without a neurodevelopmental diagnosis on the ICT-2 and GDT in terms of accuracy and confidence.

## RESULTS

Data existed for 320 participants. 18 of these individuals did not report an autism diagnosis/identify as autistic but had an elevated score on the AQ-10 (6 or above); they were excluded from the analysis as planned. In addition, 34 individuals identified as autistic but were not formally diagnosed. Our a priori analysis plan focused only on those with a formal diagnosis, so these individuals were also excluded from the hypothesis-testing set out below. Next, we excluded individuals with outlying scores on the control items of the ICT-2 or accuracy on the GDT. This included 10 people in total (3.73% of the remaining sample): 2 people with an autism diagnosis, 3 people self-reporting reading difficulties and 5 control participants. This left us with a final dataset of 260 people: 102 in Group 1 (individuals with a clinical diagnosis of autism), 49 in Group 2 (individuals with self-reported reading difficulties/dyslexia) and 109 in Group 3 (control group of individuals drawn from the general population). Please see Table 2 for demographic information and Table 3 for descriptive statistics for these groups.

### Testing hypothesis 1

For this hypothesis, we focused on performance on our two key language tests (the ICT-2 and GDT) when comparing Groups 1 and 3 (our autistic and control groups). Please see Table 4 which shows effect sizes (Cohen's *d*) for group differences on the four variables drawn from these tests. We can see that there were significant differences on the pragmatic language test (ICT-2) but not the core language test (GDT), with the autistic group showing somewhat lower scores on the ICT-2 but not the GDT. We checked whether

Wilson and Bishop (2022), *PeerJ*, DOI 10.7717/peerj.13110

**Table 1  Planned analyses.**

| Research question | Hypothesis | Statistical analysis | Power analysis |
|---|---|---|---|
| Do autistic people show reduced confidence in understanding implied meanings in conversation? | Autistic adults will score lower on the Implicature Comprehension Test-2 when responses are coded in terms of confidence (number of yes and no responses, regardless of polarity) than when responses are coded in terms of accuracy (with yes and maybe yes, and maybe no and no responses, combined according to polarity), compared to adults without any neurodevelopmental diagnosis, but will not show this same disparity between accuracy and confidence on the Grammaticality Decision Test. | A mixed model will be run including the following effects: task (Implicature Comprehension Test-2 or Grammaticality Decision Test), group (autistic or no neurodevelopmental diagnosis), and response (confidence or accuracy) as fixed effects; the interactions between these fixed effects; and participant as a random effect. The significance level of the three-way interaction will offer a test of the hypothesis. | A sample of 200 people is powered at over 98% to detect the three-way interaction. |
| Do individual differences in confidence in interpreting meaning, intolerance of uncertainty and self-rated communication difficulties intercorrelate? | The number of less confident responses (maybe responses) on the Implicature Comprehension Test-2, the score on the Intolerance of Uncertainty Scale, and self-reported social communication difficulties on the CC-SR will significantly intercorrelate across the full sample. | Pearson's correlations will be computed to quantify the relationships between these three variables across the whole sample. | A sample of 200 people is powered at over 99% to detect correlations of .3. |

**Table 2 Demographic information.**

| | Group 1, Autistic (N = 102) N | Group 2, Reading difficulties (N = 49) N | Group 3, control (N = 109) N |
|---|---|---|---|
| Women | 52 | 33 | 79 |
| Men | 43 | 15 | 30 |
| Non-binary people | 7 | 1 | 0 |
| White | 83 | 41 | 73 |
| Mixed | 5 | 3 | 7 |
| Asian | 2 | 0 | 10 |
| Black | 2 | 2 | 7 |
| Did not declare | 10 | 3 | 11 |
| Completed a degree | 62 | 28 | 73 |
| Studying for a degree | 10 | 2 | 10 |
| ADHD | 19 | 4 | 0 |
| Dyspraxia/DCD | 8 | 4 | 0 |
| Language disorder | 2 | 1 | 0 |
| Dyslexia | 11 | 25 | 0 |

| | Mean | SD | Mean | SD | Mean | SD |
|---|---|---|---|---|---|---|
| Age | 41.59 | 14.20 | 34.61 | 15.51 | 38.07 | 14.43 |

**Table 3 Descriptive Statistics.** Higher scores on the AQ-10, CC-SR, ARQ and IUS-12 indicate higher levels of the particular trait/difficulty. Higher scores on the ICAR, Synonyms Test, ICT-2 and GDT indicate stronger performance on these cognitive/language measures.

| | Group 1, Autistic (N = 102[a]) | | Group 2, Reading difficulties (N = 49[b]) | | Group 3, Control (N = 109[c]) | |
|---|---|---|---|---|---|---|
| | Mean | SD | Mean | SD | Mean | SD |
| AQ-10 total | 7.72 | 2.17 | 3.31 | 1.34 | 2.30 | 1.44 |
| CC-SR pragmatic Z-score | −2.33 | 1.74 | −0.67 | 1.26 | −0.04 | 1.16 |
| ARQ reading total | 4.25 | 3.02 | 7.94 | 2.43 | 2.27 | 1.47 |
| ICAR Total | 9.01 | 3.83 | 6.45 | 3.48 | 8.43 | 3.48 |
| Synonyms total | 15.76 | 5.17 | 11.47 | 4.87 | 13.39 | 4.74 |
| IUS-12 total | 33.25 | 9.21 | 23.86 | 7.74 | 22.39 | 9.23 |
| ICT-2 accuracy total | 35.75 | 4.15 | 37.02 | 1.90 | 37.27 | 2.27 |
| ICT-2 confidence total | 18.69 | 9.77 | 23.41 | 11.35 | 22.77 | 9.44 |
| GDT accuracy total | 43.64 | 4.44 | 41.71 | 3.86 | 43.34 | 4.92 |
| GDT confidence total | 46.17 | 4.28 | 45.55 | 4.81 | 46.59 | 3.96 |

Notes.
[a] N = 98 for CC-SR.
[b] N = 47 for CC-SR.
[c] N = 106 for CC-SR.

these differences on the ICT-2 remained when controlling for gender. We included Group (autistic vs control) and Gender (female vs not female) as factors in a two-way ANOVA for each variable extracted from the ICT-2. There remained a significant effect of Group for both accuracy, $F(1, 208) = 8.05$, $p = .005$, partial $\eta^2 = 0.04$, and confidence,

**Table 4  Cohen's $d$ for difference in performance between the autistic and control groups.** Negative differences indicate lower scores in the autistic group.

| | Cohen's $d$ | 95% CI |
|---|---|---|
| ICT-2 accuracy total | −0.46 | −0.73, −0.18 |
| ICT-2 confidence total | −0.43 | −0.70, −0.15 |
| GDT accuracy total | 0.06 | −0.21, 0.33 |
| GDT confidence total | −0.10 | −0.37, 0.17 |

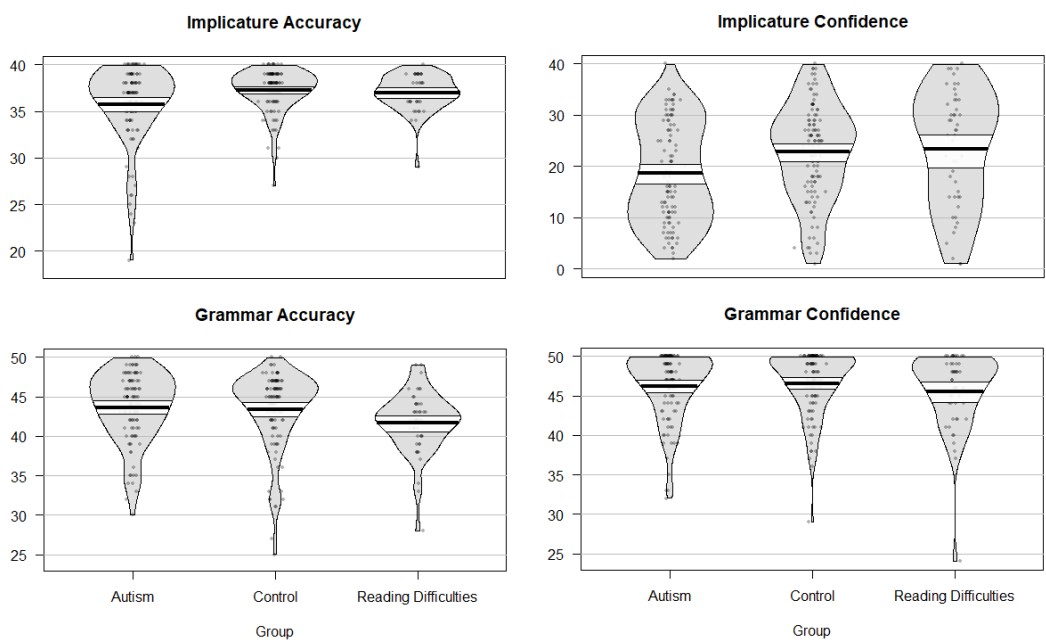

**Figure 1  Plots showing performance of the three groups on the ICT-2 and GDT.**

$F(1, 208) = 9.37$, $p = .003$, partial $\eta^2 = 0.05$, whereas Gender did not have a significant effect for either variable.

We hypothesised that the autistic group would show lower confidence on the pragmatic language test, controlling for their accuracy on the test and their performance on the core language test. We ran a mixed model with three fixed effects for Group (autistic vs control), Task (ICT-2 vs GDT), and Response (Accuracy vs Confidence) to test this hypothesis. The three-way interaction was not significant, $p = .235$, indicating that the hypothesis was not supported. As we can see from the effect sizes in Table 4, there is no evidence that the autistic group showed substantially lower confidence on the ICT-2 in relation to accuracy on the test. In fact, the autistic group gave responses on the ICT-2 that were both somewhat less accurate and less confident. They were in line with the control sample in their performance on the core language test (the GDT). See plots in Fig. 1 which show all data for each variable extracted from these tests.

In addition to this pre-registered analysis, we also looked at performance of the group with self-reported reading difficulties compared to the control group on the two tests.

**Table 5** Correlations between the variables extracted from the ICT-2 and the GDT, and self-reported communication difficulties on the CC-SR and intolerance of uncertainty on the IUS-12.

|  | CC-SR pragmatic Z-score | IUS-12 total |
| --- | --- | --- |
| IUS-12 total | −0.26 |  |
| ICT-2 accuracy total | 0.34 | −0.24 |
| ICT-2 confidence total | 0.25 | −0.25 |
| GDT accuracy total | 0.19 | −0.05 |
| GDT confidence total | 0.13 | −0.16 |

Overall, this group seemed closely comparable to the control group. The only variable extracted from the ICT-2 and GDT that showed a significant difference was Accuracy on the GDT, *i.e.*, grammatical ability. The group with self-reported reading difficulties had significantly lower scores, $t(116.16) = 2.24$, $p = 0.027$, $d = -0.35$.

### Testing hypothesis 2

Please see Table 5 for correlations between variables taken from the ICT-2 and GDT and self-reported communication difficulties on the pragmatic scale of the CC-SR and intolerance of uncertainty as measured by the IUS-12. We hypothesised that confidence on the ICT-2 and scores on the CC-SR and IUS-12 would significantly inter-correlate across the sample. This was the case, $p < .001$. Interestingly, accuracy on the ICT-2 showed similar correlations with these variables too, $p < .001$.

## DISCUSSION

We set out to replicate our previous research that found evidence for a specific difficulty with inferring implied meaning in conversation among autistic people (*Wilson & Bishop, 2020c*). As expected, autistic people were more likely than non-autistic people to give less confident and non-normative (or "inaccurate") responses on the Implicature Comprehension Test, suggesting that neurotypical and autistic people form inferences in conversation in different ways. This was in the context of no significant differences in performance on a grammar test presented in a similar format. Our findings therefore support the view that comprehension of non-literal, implied language can be a specific area of difficulty in autism even where core language skills, such as grammar and vocabulary skills, develop typically (*Andrés-Roqueta & Katsos, 2017*; *Baron-Cohen, 1988*). Likewise, our findings support the body of literature about comprehension of figurative language and nonliteral language. Although it has been suggested that differences between autistic and non-autistic people on this may be attributable to core language skills (*Kalandadze et al., 2018*), two meta-analyses have found differences between autistic and non-autistic people in understanding metaphor, proverbs and idioms, even when matching participants on verbal ability (*Morsanyi, Stamenković & Holyoak, 2020*; *Morsanyi & Stamenković, 2021*). This supports the idea that factors other than language skills are involved in the challenges experienced by autistic people in processing nonliteral language, for instance the need to integrate multiple sources of information (*Vulchanova et al., 2015*).
However, it is important to note that the effect size on the Implicature Comprehension Test was relatively small. In absolute terms, the average autistic person only gave a non-normative response one or two more times than the average non-autistic person. A few people were particularly likely to respond "inaccurately" (as shown by the long tail in the distribution for accuracy in the autistic group) but they do not represent the average person. This suggests that autistic people are not so much characterised by an "impairment" in understanding implied meaning, but may on occasion process it in such a way that leads to an incorrect response. Other studies have found that although autistic people may frequently provide accurate responses when processing metaphors, verbal analogies and inferences, they may nonetheless show increased processing time and demonstrate different strategies (*Micai et al., 2017*; *Morsanyi et al., 2021*). It is plausible that these processing differences may impact on task accuracy depending, for instance, on whether the response format of the task is more complex. Consistent with this, *Morsanyi, Stamenković & Holyoak (2020)* found that autistic people obtained lower scores on metaphor comprehension tasks when a verbal response requiring metalinguistic understanding was required rather than a nonverbal response.

In addition, we did not find the substantial difference in confidence that we expected. We hypothesised that pragmatic difficulties would be driven by much reduced confidence in processing implied meaning, whereas accuracy would only be somewhat reduced. This hypothesis was influenced by our previous study that found that autistic people gave "don't know" responses when asked about the presence of an implied meaning much more frequently than non-autistic people (*Wilson & Bishop, 2020c*). In the present study, we aimed to optimise the response format so we could measure accuracy and confidence at the same time. However, this response format gave only a modest difference in confidence between groups. In both groups, the average person only gave a confident response to half the items. While the autistic group reported somewhat lower confidence in interpreting implied meaning overall, it was common for both groups to report uncertainty, perhaps because implicature intrinsically is uncertain and statements could mean different things in different contexts. In addition, there was considerable variability in confidence ratings across both groups. This suggests that people generally vary in how sensitive they are to uncertainty in conversation and/or how prepared they are to report feeling uncertain.

Our second hypothesis linked reduced confidence on the ICT-2 to IU and self-reported communication difficulties on the CC-SR. There were significant correlations (at around the 0.3 level) between these variables, supporting the hypothesis. It seems that if an individual feels uncomfortable with uncertainty, they are more likely to report general communication troubles (as measured by the CC-SR questionnaire) and also more likely to have difficulties with making inferences in conversation. Previous studies have linked IU to various characteristics of autism, including social difficulties, sensory sensitivities, insistence on sameness and repetitive behaviours (*Hwang et al., 2020*; *Vasa et al., 2018*; *Wigham et al., 2015*). This study extends these findings to link IU to communication and pragmatic language difficulties as well. Notably, IU related to both accuracy and confidence on the ICT-2. This suggests that discomfort with uncertainty impacts not only the subjective experience of interpreting uncertain information in conversation (*i.e.*, confidence) but also

the outcome of pragmatic processing (*i.e.*, whether the person arrives at the normative interpretation). In addition to IU, it is also worth noting that the ICT-2 variables did correlate with self-reported communication difficulties (on the CC-SR), suggesting that reduced accuracy and confidence the ICT-2 does relate in a meaningful way to real-life communication difficulties.

As part of our study, we recruited a group of adults reporting reading difficulties. This group performed similarly to the control group on the Implicature Comprehension Test, suggesting that both groups processed inferences in conversation similarly. On the grammar test, there was a subtle difference in terms of accuracy. The group with reading difficulties seemed less sensitive to grammaticality compared to the control group, although both groups responded to the test with a similar degree of confidence. In this sense, the group with self-reported reading difficulties demonstrated an inverse pattern to the autistic group. While the autistic group showed some cognitive differences in their processing of implied meaning in conversation and showed intact grammatical awareness, the group with self-reported reading difficulties showed some difficulties with grammar (*i.e.*, an aspect of core language processing) but no differences in pragmatic processing.

There are a couple of issues to bear in mind when interpreting this study. The two key tests used in this study were both meta-linguistic in nature. In the ICT-2, participants were responding to comprehension questions about an exchange as a third party, rather than directly responding to implicatures directed at them. It is unclear whether the same skills are required for both these activities. It should be noted that ICT-2 scores did correlate with scores on the pragmatic scale of the CC-SR, which suggests that the ICT-2 does pick up problems relevant to day-to-day conversation. The GDT was also meta-linguistic in nature, as participants needed to make grammatical judgements about stimuli. Again, this test may be assessing skills somewhat distinct from those required in comprehension and production of grammatical speech in day-to-day contexts. The advantage of using a meta-linguistic task to measure grammatical ability in this study was that it was closely matched to the ICT-2 in terms of its response format and task demands. This meant that we could be reasonably confident in inferring that reduced performance on the ICT-2 compared to the GDT was due to some issue with processing non-literal, implied language.

In addition, there were possible limitations with the sample recruited in this study. As it was fully online, we could not validate the clinical presentation of the autistic group and relied on self-report. In addition, the autistic sample was not representative of the autistic population at large. Whereas autism is diagnosed approximately three times more often in males, over half our sample was female. However, gender did not affect performance on the ICT-2, so the gender balance in the sample may not be a significant issue. In addition, participants often had advanced education, with almost three-quarters of autistic adults either having or studying for a degree. This is not representative of the broader autistic population, which includes many individuals with co-occurring learning difficulties/disabilities. The pattern of results reported in this study may not apply to individuals with learning difficulties/disabilities, who may be more likely to have general challenges with using and understanding language rather than a specific difficulty with pragmatics.

In this registered report, we tested whether the communication difficulties in autism are linked to problems in forming confident, normative inferences during conversation. Our results supported this hypothesis, as autistic adults showed some subtle differences in performance on the Implicature Comprehension Test. We did hypothesise that reduced *confidence* in making inferences would be the key differentiator between autistic and non-autistic people. However, this was not the case, as *both* accuracy and confidence were moderately reduced on the Implicature Comprehension Test. Autistic people did not differ from non-autistic people in terms of their confidence or accuracy on a grammar test, indicating that differences were specific to inference-making rather than a more general language difficulty. By contrast, a dyslexic group showed reduced accuracy on the grammar test, but did not differ from the control group in terms of their inference-making, demonstrating a reverse pattern to the autistic group. Lastly, we found that individuals who reported discomfort with uncertainty were less likely to make confident, normative inferences in conversation and more likely to report more day-day communication difficulties. This adds to the wealth of literature around IU and autism, by demonstrating that discomfort with uncertainty also plays a role in pragmatic language and communication challenges in autism.

### Funding
This work was funded by the European Research Council (Ref 694189). The funders had no role in study design, data collection and analysis, decision to publish, or preparation of the manuscript.

### Grant Disclosures
The following grant information was disclosed by the authors:
European Research Council: 694189.

### Competing Interests
Dorothy VM Bishop is an Academic Editor for PeerJ.

### Author Contributions
- Alexander C. Wilson conceived and designed the experiments, performed the experiments, analyzed the data, prepared figures and/or tables, authored or reviewed drafts of the paper, and approved the final draft.
- Dorothy V.M. Bishop conceived and designed the experiments, authored or reviewed drafts of the paper, and approved the final draft.

### Human Ethics
The following information was supplied relating to ethical approvals (*i.e.*, approving body and any reference numbers):

Ethical approval for this project was granted on 30/03/2020 by the Medical Science Interdivisional Research Ethics Committee at Oxford University (Ref: R68518/RE001).

## Data Availability

Scripts, data and example materials for this project are available at Open Science Framework: Wilson, Alexander, and Dorothy V M Bishop. 2021. "Assessing the Confidence in Pragmatic Processing Hypothesis." OSF. June 28. doi:10.17605/OSF.IO/WK97S.

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
