# Peer review of "Stage 2 registered report: investigating a preference for certainty in conversation among autistic adults"

_PeerJ, doi:10.7717/peerj.13110_

## Round 0.1 · original submission · Minor Revisions

Both reviewers agree on the quality and benefit of this manuscript, but suggest a few minor revisions. When undertaking these revisions, please consider

- As this is a Stage 2 Registered Report, no revisions of the introduction, methods, or results are required, unless to enhance clarity where appropriate (see comments by both reviewers)

- Both reviewers note limitations which should be acknowledged in the Discussion.

- For coherence, please change the language in the Methods to past tense.

·

Basic reporting

This is an excellent and original study, which abides by the standards of pre-registration and data transparency. The topic is timely and the planned analyses are adequate.

I do have some minor suggestions, but most of them are rather optional.

First, there is no particular reason to focus the introduction on Relevance theory. Uncertainty is inherent in any model of pragmatic inferencing; on any notion of pragmatic inference or implicature, the inference at hand is non-monotonic. As far as I can see, the authors' study does not test any hypothesis which would be specific to Relevance theory as opposed to other pragmatic theories. The focus on Relevance theory may also render the introduction less friendly to people in the field of autism or clinical linguistics, who are not necessarily familiar with the details of pragmatic theorising.

Second, since the authors' hypothesis mainly bears on (un)certainty as a source of pragmatic performance in autistic individuals, they may want to test, as an additional analysis, the potential effect of trial (ideally, trial by participant random slopes should included within the models). (Un)certainty may increase during the task, and group differences could be more visible towards the end of the experiment. In the same vein, it is possible that some autistic individuals use compensatory, more explicit but also more cognitively costly, processing routes, which may, again, have an effect on response along the experiment. (In a traditional experimental setting, such an effect might show up in RTs; my guess is that owing to the online nature of this study, the RTs are rather unreliable. Is this so?)

Third, I'd use violin plots of per-participant average confidence, with superimposed fitted CIs; this would make the individual variation clearer.

Fourth, why not testing accuracy on Grammar and Implicatures tasks using logistic multilevel regressions? This would make the results much more homogeneous and easier to process.

Fourth, in testing Hypothesis 1, the authors should explain explicitly how Group, Task and Response are coded (I assume that they all are dummy coded, but this should be stated). Also, it could be worth running a stepwise, forward or backward model comparison, to clearly determine the contribution of each factor and their interactions to the model fit. Also, I'm not sure I understood why Group in this model is a binomial variable, and does not include participants with reading difficulties.

Fifth, why not including by item random intercepts?

Experimental design

The experimental design is sound. One limitation, but one that is also present in the near totality of the experimental literature on pragmatics in autism, is that the task is meta-linguistic. That is, the authors test how participants understand an exchange as a third party, not whether and how they process an implicature directed at them. This point may be worth discussing in generalising the results to communication difficulties autistic individuals may face in real life.

Validity of the findings

Findings are valid and valuable. See the comment above on result generalisation.

·

Basic reporting

The level of presenting the study is overall clear throughout the text. There are, however, a few exceptions. For instance, the abstract fails to convey the findings in a clear and unambiguous way. References to the literature are insufficient and quite limited. There has been a wealth of recent studies (including experimental studies, reviews and meta-analyses) of figurative/non-literal language skills in autism and the evidence is still controversial regarding exactly what might be the main contributing factor in the well-observed problems with non-literal language comprehension on the spectrum, in particular, to what extent structural language skills explain the variance. in this respect, two recent reviews and meta-analyses by Morsanyi & colleagues are worth reviewing for a more balanced picture. The early review by Loukusa & Moilanen (2009) is misrepresented as providing evidence in support of the language problem account. in that study "pragmatic inference weaknesses, but not inabilities, were found throughout the studies. However, researchers did not wholly agree on the reasons and the extent of processing difficulties. The most commonly suggested explanations for pragmatic inference deficits were theory of mind and central coherence."
Recent research on inferencing ability in highly-verbal individuals with autism (Micai & colleagues) suggests also problems with meta-cognitive skills and not clearly differentiating between goals in individuals with autism in comparison to age- and language-matched controls. This may be an additional source of variance for the communication problems. As reflected in these studies, and many other recent studies, the problems for language-matched individuals on the spectrum do not reside as much in lower accuracy. Rather, they are reflected in processing speed and employing different/compensatory strategies.
The structure of the paper is appropriate and the figures and tables are informative.

Experimental design

The paper reports original primary research and is well within the scope and aims of the journal. The research question is well defined, relevant and original and clearly deserves empirical support (as planned for the study). The study is marked by overall rigorous design and consistent with the research question and hypotheses.
Even though the measures collected for the analyses are well described, their role in the analyses is not sufficiently well identified. For instance, results from the ICAR are not reported and not taken into account in the analyses. The same applies to autistic traits/symptomatology as measured on the AQ-10. Often in studies of autism, the latter are included as predictors of performance.
The use of a grammaticality judgement test may be problematic when assessing structural language competence, simply because it is a meta-linguistic task. This deserves being addressed.
What was the role of the 16 explicit response items? How were they analysed? This is probably an overall issue with the design, namely that it did not include an equal number of stimuli requiring explicit, literal interpretation.

Validity of the findings

The results are adequately reported and addressed in the discussion. It also deserves merit reporting and discussing results which do not support the original hypotheses. More in-depth discussion is in order, especially given the gaps identified in introducing the background and context of the study.

Additional comments

This is an interesting and important paper which deserves publication after the critical issues have been addressed.

---

## Round 0.2 · accepted · Accept

I have reviewed the changes that were made in response to the reviewers' comments, and determined that all outstanding issues have been addressed. Many thanks for the detailed rebuttal report, and congratulations on the publications on this interesting and well-described study!